# Kernel-based gene–environment interaction tests for rare variants with multiple quantitative phenotypes

**Xiaoqin Jin, Gang Shi** [ID]*

State Key Laboratory of Integrated Services Networks, Xidian University, Xi'an, Shaanxi, China

* gshi@xidian.edu.cn

**Data Availability Statement:** This research has been conducted using the UK Biobank Resource under Application Number 44080. The genetic and phenotype datasets are available via the UK

## Abstract

Previous studies have suggested that gene–environment interactions (GEIs) between a common variant and an environmental factor can influence multiple correlated phenotypes simultaneously, that is, GEI pleiotropy, and that analyzing multiple phenotypes jointly is more powerful than analyzing phenotypes separately by using single-phenotype GEI tests. Methods to test the GEI for rare variants with multiple phenotypes are, however, lacking. In our work, we model the correlation among the GEI effects of a variant on multiple quantitative phenotypes through four kernels and propose four multiphenotype GEI tests for rare variants, which are a test with a homogeneous kernel (Hom-GEI), a test with a heterogeneous kernel (Het-GEI), a test with a projection phenotype kernel (PPK-GEI) and a test with a linear phenotype kernel (LPK-GEI). Through numerical simulations, we show that correlation among phenotypes can enhance the statistical power except for LPK-GEI, which simply combines statistics from single-phenotype GEI tests and ignores the phenotypic correlations. Among almost all considered scenarios, Het-GEI and PPK-GEI are more powerful than Hom-GEI and LPK-GEI. We apply Het-GEI and PPK-GEI in the genome-wide GEI analysis of systolic blood pressure (SBP) and diastolic blood pressure (DBP) in the UK Biobank. We analyze 18,101 genes and find that *LEUTX* is associated with SBP and DBP ($p = 2.20 \times 10^{-6}$) through its interaction with hemoglobin. The single-phenotype GEI test and our multiphenotype GEI tests Het-GEI and PPK-GEI are also used to evaluate the gene–hemoglobin interactions for 22 genes that were previously reported to be associated with SBP or DBP in a meta-analysis of genetic main effects. *MYO1C* shows nominal significance ($p < 0.05$) by the Het-GEI test. *NOS3* shows nominal significance in DBP and *MYO1C* in both SBP and DBP by the single-phenotype GEI test.

## Introduction

Genome-wide association studies (GWASs) have identified numerous common variants associated with common diseases or phenotypes [1]. Nevertheless, a small portion of the heritabilities can be explained by the discovered common variants [2, 3]. Sequencing studies showed that some of the "missing heritability" was attributable to rare variants [4, 5]. Complex diseases

Biobank data access process. More details are available at http://www.ukbiobank.ac.uk/register-apply/.

**Funding:** This work was supported by the national Thousand Youth Talents Plan to GS. The funders had no role in study design, data collection and analysis, decision to publish, or preparation of the manuscript.

**Competing interests:** The authors have declared that no competing interests exist.

are usually influenced by genetic factors, environmental factors and the interplay between them. Wang et al. showed that the interactions between *SMC5* variants and alcohol consumption are associated with fasting plasma lipid levels [6]. Yang et al. demonstrated that the interactions between *PDE3B* variants and smoking are associated with pulmonary function [7]. Johansson et al. revealed that the interactions between *NFE2L2* variants and second-hand smoke are associated with pediatric asthma risk [8]. For a long time, gene–environment interactions (GEIs) have been expected to explain some of the "missing heritability" and shed light on the genetic etiology of complex diseases [9].

Existing studies suggest that the interaction between a common variant and an environmental factor may be associated with multiple correlated phenotypes, which is called GEI pleiotropy [10]. Kilpeläinen et al. identified four loci in or near *CLASP1*, *LHX1*, *SNTA1* and *CNTNAP2* that are associated with three blood lipid levels: low density lipoprotein, high density lipoprotein and triglycerides through their interactions with physical activity [11]. Novel gene-sleep interactions were also identified for known lipid loci, including *LPL* and *PCSK9* [12]. To date, all the reported GEI pleiotropies are with common variants. From a methodological perspective, Majumdar et al. showed that statistical power to detect GEI effects can be improved by analyzing multiple phenotypes jointly [10]. However, multiphenotype methods for testing GEIs with rare variants are lacking.

To the best of our knowledge, there is only one method currently available for testing GEIs with rare variants and multiple phenotypes [13]. The method consists of three steps: remove correlation among multiple phenotypes by using principal component analysis or other linear transformations; obtain p value for each transformed phenotype by testing the effects of an optimally weighted combination of GEIs for rare variants (TOW-GE) [14]; employ Fisher's combination test (FCT) to combine the p values of multiple phenotypes. We denote the method as TOWGE-FCT in this paper. It can be expected that the degree of freedom of TOW-GE-FCT would become larger with the increasing number of phenotypes, which might limit statistical power of the test.

In this work, we model the correlations among the GEI effects of a variant on multiple phenotypes by assuming four different kernel matrices, similar to those for multiphenotype tests of genetic main effects [15]. We extend the single-phenotype GEI test [16] and propose four multiphenotype GEI tests for rare variants, which are the test with homogeneous kernel (Hom-GEI), the test with heterogeneous kernel (Het-GEI), the test with projection phenotype kernel (PPK-GEI) and the test with linear phenotype kernel (LPK-GEI). We conduct simulation studies to examine the empirical distributions of the four test statistics under the null hypothesis and compare their statistical power under different scenarios. In the analysis of systolic blood pressure (SBP) and diastolic blood pressure (DBP) in the UK Biobank, we chose hemoglobin (Hb) as the environmental variable, which is known to be associated with both SBP and DBP [17, 18]. With the whole-exome sequencing data in 200,643 samples, we applied Het-GEI and PPK-GEI in the genome-wide analyses of gene-Hb interactions. We also carried out single-phenotype and multiphenotype GEI tests to evaluate the gene-Hb interactions for 22 genes that were previously reported to be associated with SBP or DBP in a meta-analysis of main genetic effects [19].

## Methods

### Single-phenotype GEI test

Assume that $n$ unrelated individuals are sequenced in a gene or region with $m$ rare variants and $K$ quantitative phenotypes are measured. For the $k$-th phenotype, $\boldsymbol{y}_k = (y_{1k}, y_{1k}, \cdots, y_{nk})^{\mathrm{T}}$ denotes an $n \times 1$ phenotype vector, and $\boldsymbol{X} = (\boldsymbol{X}_1, \boldsymbol{X}_2, \cdots, \boldsymbol{X}_{q+1})$ is an $n \times (q + 1)$ matrix

comprised of intercept and covariate vectors with $X_t = (X_{1t}, X_{2t}, \cdots, X_{nt})^{\mathrm{T}}$, $t = 1, 2, \ldots, q+1$. The first vector $X_1$ represents the intercept vector with elements $X_{i1} = 1$ and $i = 1, 2, \cdots, n$. The other $q$ vectors are the covariate vectors. Let $G = (G_1, G_1, \cdots, G_m)$ be an $n \times m$ genotype matrix, in which $G_j = (G_{1j}, G_{2j}, \cdots, G_{nj})^{\mathrm{T}}$, $j = 1, 2, \ldots, m$, and $G_{ij}$ is the number of minor alleles. $E = \mathrm{diag}\{E_i\}$ denotes an $n \times n$ diagonal matrix of environmental measurements, and $E_i$ is centralized and included in $X$ as a covariate for adjusting the environmental effect. Following the single-phenotype GEI test for rare variants in rareGE [16], we consider the linear mixed model as follows:

$$y_k = X\alpha_k + GW\beta_k + EGW\gamma_k + \varepsilon_k, \tag{1}$$

where $\alpha_k = (\alpha_{k1}, \alpha_{k2}, \cdots, \alpha_{k(q+1)})^{\mathrm{T}}$ is a $(q+1) \times 1$ vector of covariate effects for the $k$-th phenotype. $W = \mathrm{diag}\{w_j\}$ is an $m \times m$ weight matrix for the $m$ variants. The weight of the $j$-th variants is $w_j = \mathrm{Beta}(\mathrm{MAF}_j, 1, 25)$ [20], where $\mathrm{MAF}_j$ is the minor allele frequency (MAF) of the $j$-th variants. In addition, $\beta_k = (\beta_{1k}, \beta_{2k}, \cdots, \beta_{mk})^{\mathrm{T}}$ is an $m \times 1$ vector consisting of genetic main effects for the $k$-th phenotype, and $\gamma_k = (\gamma_{1k}, \gamma_{2k}, \cdots, \gamma_{mk})^{\mathrm{T}}$ is an $m \times 1$ vector of the interaction effects. Here, the main genetic effects $\beta_k$ are assumed to be fixed and the interaction effects $\gamma_k$ to be random, $\gamma_k \sim \mathrm{MVN}(0, \sigma^2 I_m)$. In addition, $\varepsilon_k = (\varepsilon_{1k}, \varepsilon_{2k}, \cdots, \varepsilon_{nk})^{\mathrm{T}}$ denotes an $n \times 1$ error vector, and $\varepsilon_k \sim MVN(0, \sigma_k^2 I_n)$. The null hypothesis for testing the GEI interactions is $\mathrm{H}_0: \sigma^2 = 0$. The model under the null hypothesis is

$$y_k = X\alpha_k + GW\beta_k + \varepsilon_k, \tag{2}$$

Here, $\alpha_k$, $\beta_k$, and $\sigma_k^2$ can be estimated by linear regression, and the estimated mean and variance-covariance matrix of $y_k$ are

$$\hat{\mu}_k = X\hat{\alpha}_k + GW\hat{\beta}_k$$

$$\hat{V}_k = \hat{\sigma}_k^2 I_n$$

where $\hat{\mu}_k = (\hat{\mu}_{1k}, \hat{\mu}_{2k}, \cdots, \hat{\mu}_{nk})^T$, $(\hat{\alpha}_k^T, \hat{\beta}_k^T)^T = (Z^T Z)^{-1} Z^T y_k$ with $Z = (X, GW)$. The score statistic for testing the GEI effects is

$$Q_k = (y_k - \hat{\mu}_k)^{\mathrm{T}} \hat{V}_k^{-1} EGWWG^{\mathrm{T}} E \hat{V}_k^{-1} (y_k - \hat{\mu}_k), \tag{3}$$

which is mathematically equivalent to

$$Q_k = \sum_{j=1}^{m} w_j^2 S_{jk}^2. \tag{4}$$

Here, $S_{jk} = \sum_{i=1}^{n} E_i G_{ij} (y_{ik} - \hat{\mu}_{ik}) / \hat{\sigma}_k^2$ is the score statistic for the $j$-th variant.

Under $\mathrm{H}_0$, $Q_k \sim \sum_j \lambda_j \chi_{1,j}^2$ follows a mixture of chi-square distributions with 1 degree of freedom, and $\lambda_j$ are nonzero eigenvalues of the regional relationship matrix

$$\Psi_k = WG^{\mathrm{T}} E(I_n - Z(Z^{\mathrm{T}}Z)^{-1}Z^{\mathrm{T}}) \hat{V}_k^{-1} EGW. \tag{5}$$

The p-value can be computed by using Kuonen's saddlepoint approximation method [16, 21]. In the same spirit, we extend the single-phenotype GEI test for multiple phenotypes.

## Kernel-based multiphenotype GEI tests

Denote $y = (y_1, y_2, \cdots, y_K)$ as the $n \times K$ matrix of $K$ phenotypes and $A = (\alpha_1, \alpha_2, \cdots, \alpha_K)$ as the $(q+1) \times K$ matrix of covariate effects. Let $B = (\beta_1, \beta_2, \cdots, \beta_K)$ be the $m \times K$ matrix of genetic

main effects and $\boldsymbol{\Gamma} = (\boldsymbol{\gamma}_1, \boldsymbol{\gamma}_2, \cdots, \boldsymbol{\gamma}_K)$ be the $m \times K$ matrix of GEI effects. In addition, $\boldsymbol{\varepsilon} = (\boldsymbol{\varepsilon}_1, \boldsymbol{\varepsilon}_2, \cdots, \boldsymbol{\varepsilon}_K)$ is the $n \times K$ error matrix. In light of the correlation among phenotypes, we assume $\boldsymbol{\varepsilon} = (\varepsilon_1, \varepsilon_2, \cdots, \varepsilon_K) \sim \text{MVN}(\mathbf{0}, \boldsymbol{\Sigma})$, $i = 1, 2, \cdots, n$. Then, the mixed model for multiple phenotypes can be formulated in a matrix form as follows:

$$\boldsymbol{y} = \boldsymbol{XA} + \boldsymbol{GWB} + \boldsymbol{EGW\Gamma} + \boldsymbol{\varepsilon}. \tag{6}$$

Stack columns of the phenotype matrix $\boldsymbol{y}$ into a vector $\text{vec}(\boldsymbol{y}) = (\boldsymbol{y}_1^T, \boldsymbol{y}_2^T, \cdots, \boldsymbol{y}_K^T)^T$ and columns of the error matrix $\boldsymbol{\varepsilon}$ into $\text{vec}(\boldsymbol{\varepsilon}) = (\boldsymbol{\varepsilon}_1^T, \boldsymbol{\varepsilon}_2^T, \cdots, \boldsymbol{\varepsilon}_K^T)^T$. We have $\text{vec}(\boldsymbol{\varepsilon}) \sim \text{MVN}(\mathbf{0}, \boldsymbol{\Sigma} \otimes \boldsymbol{I}_n)$, where $\otimes$ is the Kronecker product [22]. We rewrite model (6) in vector form as

$$\text{vec}(\boldsymbol{y}) = (\boldsymbol{I}_K \otimes \boldsymbol{X})\text{vec}(\boldsymbol{A}) + (\boldsymbol{I}_K \otimes \boldsymbol{GW})\text{vec}(\boldsymbol{B}) + (\boldsymbol{I}_K \otimes \boldsymbol{EGW})\text{vec}(\boldsymbol{\Gamma}) + \text{vec}(\boldsymbol{\varepsilon}). \tag{7}$$

Assume $\text{vec}(\boldsymbol{\Gamma}) \sim \text{MVN}(\mathbf{0}, \sigma^2 \boldsymbol{\Sigma}_P \otimes \boldsymbol{I}_m)$, where $\boldsymbol{\Sigma}_P$ is a $K \times K$ kernel in the phenotype space and models the correlation among the GEI effects of a variant on multiple phenotypes. As a result, $\text{vec}(\boldsymbol{y}) \sim \text{MVN}(\text{vec}(\boldsymbol{\mu}), \boldsymbol{H})$, where $\boldsymbol{\mu} = \boldsymbol{XA} + \boldsymbol{GWB}$ and $\boldsymbol{H} = \sigma^2(\boldsymbol{\Sigma}_P \otimes \boldsymbol{EGWWG}^T \boldsymbol{E}) + \boldsymbol{\Sigma} \otimes \boldsymbol{I}_n$. The null hypothesis for testing the GEI effects with multiple phenotypes is $H_0$: $\sigma^2 = 0$, and the score statistic is

$$Q = \text{vec}(\boldsymbol{y} - \hat{\boldsymbol{\mu}})^T \{ (\hat{\boldsymbol{\Sigma}}^{-1} \boldsymbol{\Sigma}_P \hat{\boldsymbol{\Sigma}}^{-1}) \otimes (\boldsymbol{EGWWG}^T \boldsymbol{E}) \} \text{vec}(\boldsymbol{y} - \hat{\boldsymbol{\mu}}), \tag{8}$$

where $\hat{\boldsymbol{\mu}}$ and $\hat{\boldsymbol{\Sigma}}$ are the estimated mean and variance-covariance matrix, respectively.

The score statistic $Q$ asymptotically follows a mixture of 1-freedom chi-square distributions $\sum_j \lambda_j \chi_{1,j}^2$ and $\lambda_j$ are nonzero eigenvalues of

$$\boldsymbol{\Psi} = \boldsymbol{\Sigma}_P^{1/2} \hat{\boldsymbol{\Sigma}}^{-1} \boldsymbol{\Sigma}_P^{1/2} \otimes \boldsymbol{WG}^T \boldsymbol{EPEGW}, \tag{9}$$

where $\boldsymbol{P} = \boldsymbol{I}_n - \boldsymbol{Z}(\boldsymbol{Z}^T\boldsymbol{Z})^{-1}\boldsymbol{Z}^T$ and $\boldsymbol{Z}$ is the same as in the single-phenotype GEI test. The corresponding p-values can be computed via Kuonen's saddlepoint method [21].

As can be seen in (8), our proposed tests depend on the kernel matrix $\boldsymbol{\Sigma}_P$. Similar to [15], we use four types of kernel matrices to model the correlation among the GEI effects on multiple phenotypes.

**Homogeneous kernel.** Assume that the GEI effects of a variant on multiple different phenotypes are homogeneous, implying that $\gamma_{j1} = \gamma_{j2} = \cdots = \gamma_{jK}$. The kernel is constructed as

$$\boldsymbol{\Sigma}_P = \boldsymbol{\Sigma}_{\text{Hom}} = \mathbf{1}_K \mathbf{1}_K^T,$$

where $\mathbf{1}_K = (1, 1, \cdots, 1)^T$ is a $K \times 1$ vector. $\boldsymbol{\Sigma}_{\text{Hom}}$ indicates the GEI effects of a variant on multiple phenotypes to be the same.

**Heterogeneous kernel.** Assuming that the GEI effect sizes of a variant on multiple phenotypes are heterogeneous, the kernel is

$$\boldsymbol{\Sigma}_P = \boldsymbol{\Sigma}_{\text{Het}} = \boldsymbol{I}_K,$$

where $\boldsymbol{I}_K$ is a $K \times K$ identity matrix. Here, $\boldsymbol{\Sigma}_{\text{Het}}$ implies that the GEI effects of a variant on multiple phenotypes are independent.

**Projection phenotype kernel.** Assume that the correlation among the GEI effects of a variant on multiple phenotypes can be depicted by the correlation among the phenotypes. That is,

$$\boldsymbol{\Sigma}_P = \boldsymbol{\Sigma}_{\text{PPK}} = \hat{\boldsymbol{\Sigma}},$$

where $\hat{\boldsymbol{\Sigma}}$ is the estimated variance-covariance matrix of the phenotypes.

**Linear phenotype kernel.** Assume that the GEI effects of a variant on multiple phenotypes equal the squared correlation among the phenotypes. That is,

$$\boldsymbol{\Sigma}_{\mathrm{P}} = \boldsymbol{\Sigma}_{\mathrm{LPK}} = \hat{\boldsymbol{\Sigma}}^2.$$

Similar to the proof in [22], the test score statistic (8) is

$$Q_{\mathrm{LPK-GEI}} = \{\mathrm{vec}(\boldsymbol{y}) - \mathrm{vec}(\hat{\boldsymbol{\mu}})\}^T \{\boldsymbol{I}_K \otimes (\boldsymbol{EGWWG}^T\boldsymbol{E})\}\{\mathrm{vec}(\boldsymbol{y}) - \mathrm{vec}(\hat{\boldsymbol{\mu}})\}, \qquad (10)$$

which can be rewritten as

$$Q_{\mathrm{LPK-GEI}} = \sum_{k=1}^{K}\sum_{j=1}^{m} w_j^2 S_{jk}^2. \qquad (11)$$

Therefore, the LPK-GEI test simply combines statistics of single-phenotype GEI tests across multiple phenotypes.

Based on different choices of the kernel matrix $\boldsymbol{\Sigma}_{\mathrm{P}}$, we propose four multiphenotype GEI tests, which are named Hom-GEI, Het-GEI, PPK-GEI and LPK-GEI.

## Results

### Numerical simulations

To evaluate the null distributions and statistical power of the four proposed tests, we carried out extensive simulation studies. Using the calibrated coalescent model implemented in COSI [23], we generated 10,000 haplotypes in a 200 kb genomic region. Parameters in the coalescent model were used to mimic the linkage disequilibrium pattern, local recombination rate and demographic history for the population of European ancestry. We randomly paired these haplotypes to form diploid genotype data of 10,000 individuals and randomly selected 5000 out of the 10,000 individuals. A subregion length of 3 kb was randomly selected from the 200 kb region to obtain the genotype data of the 5000 samples for each replicate, and 1000 replicates of genotype data were generated. Variants with MAF $\leq 0.01$ were considered to be rare and used for simulations.

To evaluate null distributions of our proposed tests, four phenotypes of the 5000 unrelated individuals under the null hypothesis were generated. For the sake of simplicity, phenotypes shared the same covariate sets and were generated as follows:

$$
\begin{aligned}
y_{i1} &= 0.1sex_i + 0.05age_i + 0.1bmi_i + \sum_{j=1}^{m} G_{ij}w_j\beta_{j1}l_1 + \varepsilon_{i1} \\
y_{i2} &= 0.5sex_i + 0.05age_i + 0.1bmi_i + \sum_{j=1}^{m} G_{ij}w_j\beta_{j2}l_2 + \varepsilon_{i2} \\
y_{i3} &= 0.1sex_i + 0.05age_i + 0.1bmi_i + \sum_{j=1}^{m} G_{ij}w_j\beta_{j3}l_3 + \varepsilon_{i3} \\
y_{i4} &= 0.5sex_i + 0.05age_i + 0.1bmi_i + \sum_{j=1}^{m} G_{ij}w_j\beta_{j4}l_4 + \varepsilon_{i4}
\end{aligned}, \qquad (12)
$$

Where $y_{i1}, y_{i2}, y_{i3}, y_{i4}$ are the four phenotypes for the $i$-th individual ($i = 1, 2, \ldots, n$). For the $i$-th individual, $sex_i$ is a binary covariate following a Bernoulli distribution with probability 0.5, namely, $sex_i \sim \mathrm{Bernoulli}(0.5)$. Both $age_i$ and $bmi_i$ are continuous covariates: $age_i \sim \mathrm{N}(50, 25)$, $bmi_i \sim \mathrm{N}(50, 25)$. $G_{ij}$ ($j = 1, 2, \cdots, m$) are the coded genotypes of the simulated causal variants

for individual $i$. Here, we assumed a proportion of causal variants $\theta = 0.1, 0.2, 0.3$. In addition, $w_j$ ($j = 1, 2, \cdots, m$) is the weight of variant $j$; $\beta_{j1}, \beta_{j2}, \beta_{j3}$, and $\beta_{j4}$ are the main genetic effects for phenotype 1, phenotype 2, phenotype 3 and phenotype 4, respectively, with $\beta_{j1} = 0.1, \beta_{j2} = 0.2$, $\beta_{j3} = 0.1$, and $\beta_{j4} = 0.2$; and $l_1, l_2, l_3$, and $l_4$ are indicator variables, with $l_k = 1$ when phenotype $k$ is associated with genetic variants and $l_k = 0$ otherwise. Since not all of the phenotypes may be associated with the rare variants [22, 24], we considered scenarios under which pleiotropy exists or does not exist. Specifically, we assumed that the first $t$ phenotypes were associated with the rare variants, namely, $l_1 = \cdots = l_t = 1$ and $l_{t+1} = \cdots = l_{4-t} = 0$, $t = 1,2,3,4$. $\varepsilon_{ik}$ is a random error for the $i$-th individual and the $k$-th phenotype, $\boldsymbol{\varepsilon}_i = (\varepsilon_{i1}, \varepsilon_{i2}, \varepsilon_{i3}, \varepsilon_{i4})^{\mathrm{T}} \sim \mathrm{MVN}(\mathbf{0}, \boldsymbol{\Sigma})$,

where $\boldsymbol{\Sigma} = \begin{bmatrix} 2 & \sqrt{2}\rho & \sqrt{2}\rho & \sqrt{2}\rho \\ \sqrt{2}\rho & 1 & \rho & \rho \\ \sqrt{2}\rho & \rho & 1 & \rho \\ \sqrt{2}\rho & \rho & \rho & 1 \end{bmatrix}$ and $\rho$ represents the correlation among different

phenotypes. Three levels of correlation strength were considered: weak correlation with $\rho = 0.25$, moderate correlation with $\rho = 0.5$ and strong correlation with $\rho = 0.75$. For each simulation setup, 20 replicates of phenotypes and covariates were simulated based on one genotype dataset; thus, a total of 20,000 replicates of phenotypes and covariates were simulated.

To evaluate the statistical power of our proposed four multiphenotype GEI tests, we simulated the four correlated phenotypes for 5000 independent individuals under the alternative hypothesis. For each of the genotype datasets, one phenotype and covariates set was simulated according to the following model:

$$y_{i1} = 0.1sex_i + 0.05age_i + 0.1bmi_i + \sum_{j=1}^{m} G_{ij}w_j\beta_{j1}l_1 + \sum_{j=1}^{m} E_iG_{ij}w_j\gamma_{j1}l_1 + \varepsilon_{i1}$$

$$y_{i2} = 0.5sex_i + 0.05age_i + 0.1bmi_i + \sum_{j=1}^{m} G_{ij}w_j\beta_{j2}l_2 + \sum_{j=1}^{m} E_iG_{ij}w_j\gamma_{j2}l_2 + \varepsilon_{i2}$$

$$y_{i3} = 0.1sex_i + 0.05age_i + 0.1bmi_i + \sum_{j=1}^{m} G_{ij}w_j\beta_{j3}l_3 + \sum_{j=1}^{m} E_iG_{ij}w_j\gamma_{j3}l_3 + \varepsilon_{i3} \tag{13}$$

$$y_{i4} = 0.5sex_i + 0.05age_i + 0.1bmi_i + \sum_{j=1}^{m} G_{ij}w_j\beta_{j4}l_4 + \sum_{j=1}^{m} E_iG_{ij}w_j\gamma_{j4}l_4 + \varepsilon_{i4}$$

where $sex_i$, $age_i$, $bmi_i$, $G_{ij}$, $w_j$, $\beta_{jk}$ ($k = 1,2,3,4$), $l_k$ ($k = 1,2,3,4$) and $\boldsymbol{\varepsilon}_i = (\varepsilon_{i1}, \varepsilon_{i2}, \varepsilon_{i3}, \varepsilon_{i4})^{\mathrm{T}}$ are the same as described in model (12). The body mass index (BMI) was centered and used as the environmental variate $E_i$. Here, $\gamma_{jk}$ is the gene-BMI interaction effect of the $j$-th causal rare variant on the $k$-th phenotype, with $\gamma_{jk} \sim N(0, 0.05)^2$. Since the interaction effects of a variant on each phenotype were simulated independently, the gene–BMI interaction effects of a variant on multiple phenotypes are heterogeneous.

In all simulations and the analyses of the simulated data, variant weights were the density function of beta distribution with degrees of freedom of 1 and 25 evaluated at the MAF of rare variants [20] as described in the single-phenotype GEI test. We considered the gene–BMI interaction to be significant if its p-value was less than $2.5 \times 10^{-6}$, corresponding to a correction for multiple testing in genome-wide studies of 20,000 genes. Empirical power was the portion of significant results in 1000 replicates.

**Null distributions.** We examined null distributions of test statistics for Hom-GEI, Het-GEI, PPK-GEI and LPK-GEI with causal rare variant proportion $\theta = 0.2$ and all phenotypes associated with genetic variants. We first estimated means and residuals by performing

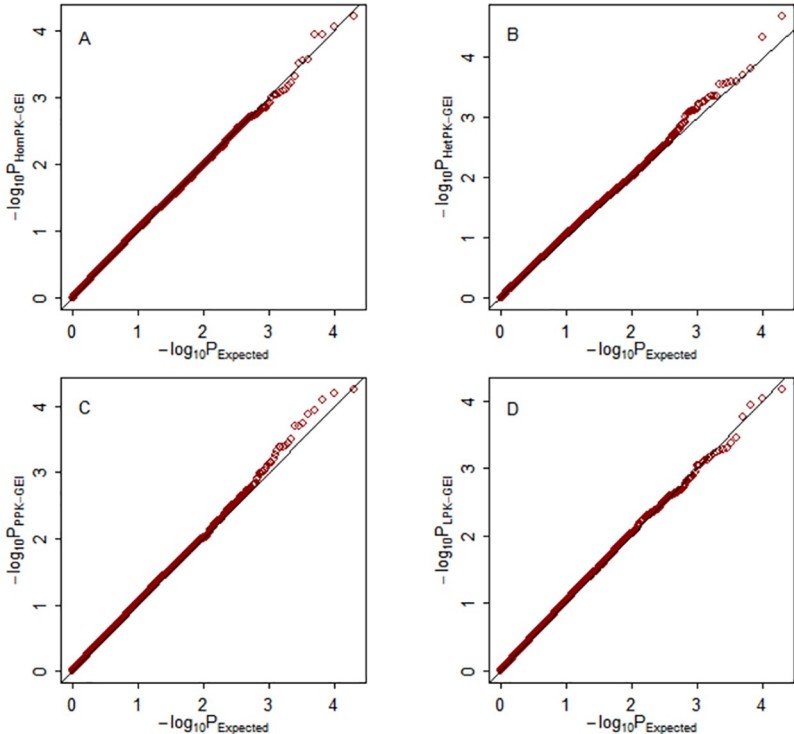

**Fig 1. Q–Q plots of the test statistics under the null hypothesis with weak among-phenotype correlation $\rho = 0.25$.**
The horizontal and vertical axes represent the negative $\log_{10}$ of the expected p-values and the negative $\log_{10}$ of the observed p-values, respectively. A: Hom-GEI; B: Het-GEI; C: PPK-GEI; D: LPK-GEI.

phenotype-specific regression analyses. The variance-covariance matrix $\Sigma$ was estimated by residuals from all phenotypes. Test statistics of the four tests were computed as in (8) with corresponding kernels for Hom-GEI, Het-GEI, PPK-GEI and LPK-GEI. Using Kuonen's saddle-point approximation method [21], the p-values of the four test statistics were computed. Finally, we compared distributions of empirical p-values with the expected uniform distribution between 0 and 1.

The quantile-quantile (Q–Q) plots of the four multiphenotype statistics under weak ($\rho = 0.25$), moderate ($\rho = 0.5$) and strong ($\rho = 0.75$) correlations among phenotypes are shown in Figs 1–3, respectively. The empirical distributions of the four test statistics are aligned with their theoretical distributions, as expected.

**Statistical power.** The statistical power of Hom-GEI, Het-GEI, PPK-GEI and LPK-GEI under weak, moderate and strong correlations among phenotypes are shown in Figs 4–6, respectively. In each figure, the power with three different proportions of causal variants and different numbers of associated phenotypes are presented. All four tests show improved power as the proportion of causal variants increases. This is because an increased proportion of causal variants leads to larger interaction effects under the test. Taking Fig 5D as an example, for the causal rare variant proportion $\theta = 0.1$, Hom-GEI, Het-GEI, PPK-GEI and LPK-GEI have powers of 0.067, 0.424, 0.431 and 0.199, respectively. For $\theta = 0.2$, the corresponding powers are 0.145, 0.72, 0.725 and 0.428, respectively. For $\theta = 0.3$, the power increases further to 0.278, 0.867, 0.869 and 0.608.

From Figs 4–6, we can see that the four tests provide improved power with more phenotypes associated with the interactions, which suggests that our multiphenotype analyses can

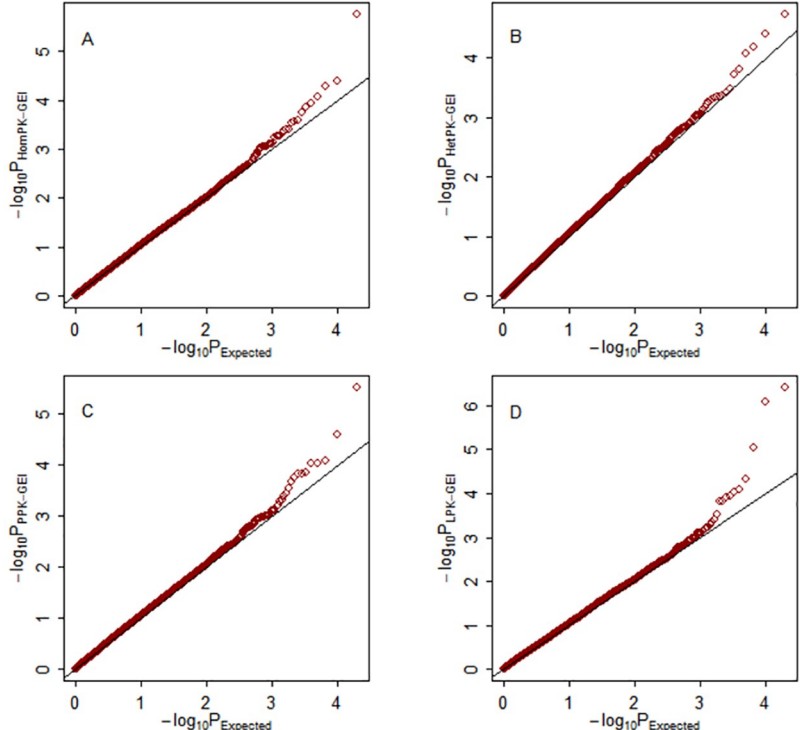

**Fig 2. Q–Q plots of the test statistics under the null hypothesis with moderate among-phenotype correlation $\rho$ = 0.5.** The horizontal and vertical axes represent the negative $\log_{10}$ of the expected p-values and the negative $\log_{10}$ of the observed p-values, respectively. A: Hom-GEI; B: Het-GEI; C: PPK-GEI; D: LPK-GEI.

exploit GEI pleiotropy effectively. For instance, in Fig 6, with the causal proportion $\theta$ = 0.2, when only the first phenotype is associated with the interactions, the powers of Hom-GEI, Het-GEI, PPK-GEI and LPK-GEI are 0.013, 0.206, 0.292 and 0.045, respectively, as shown in Fig 6A. When the first two phenotypes are associated with the interactions, the corresponding power improves to 0.092, 0.607, 0.629 and 0.152, as shown in Fig 6B. With three phenotypes associated with the interactions, the power become even larger and are 0.173, 0.766, 0.767 and 0.252, as shown in Fig 6C. When all phenotypes are associated with the interactions, the power further increase to 0.236, 0.827, 0.828 and 0.345, as shown in Fig 6D.

We can also see from Figs 4–6 that Hom-GEI, Het-GEI and PPK-GEI have enhanced power as the correlation among phenotypes becomes stronger, but LPK-GEI suffers power loss. For instance, we observe the power of the four tests with causal proportion $\theta$ = 0.2. When the correlation among phenotypes is weak, the power values of Hom-GEI, Het-GEI, PPK-GEI and LPK-GEI are 0.105, 0.508, 0.523 and 0.370, respectively, in Fig 4C. When the correlation is moderate, the power values are 0.104, 0.597, 0.598 and 0.311, as shown in Fig 5C. With a strong correlation, the power values are 0.173, 0.766, 0.767 and 0.252, as shown in Fig 6C. This demonstrates that Hom-GEI, Het-GEI and PPK-GEI can benefit from the increased correlations among phenotypes. However, since LPK-GEI directly combines statistics from single-phenotype GEI tests and ignores the phenotypic correlations, the increased correlation among phenotypes leads to a substantial power loss.

Among almost all of the considered scenarios in Figs 4–6, PPK-GEI has approximately the same or slightly larger power than Het-GEI, and the two tests outperform Hom-GEI and LPK-GEI. Hom-GEI shows the poorest power performance among all the proposed tests. This

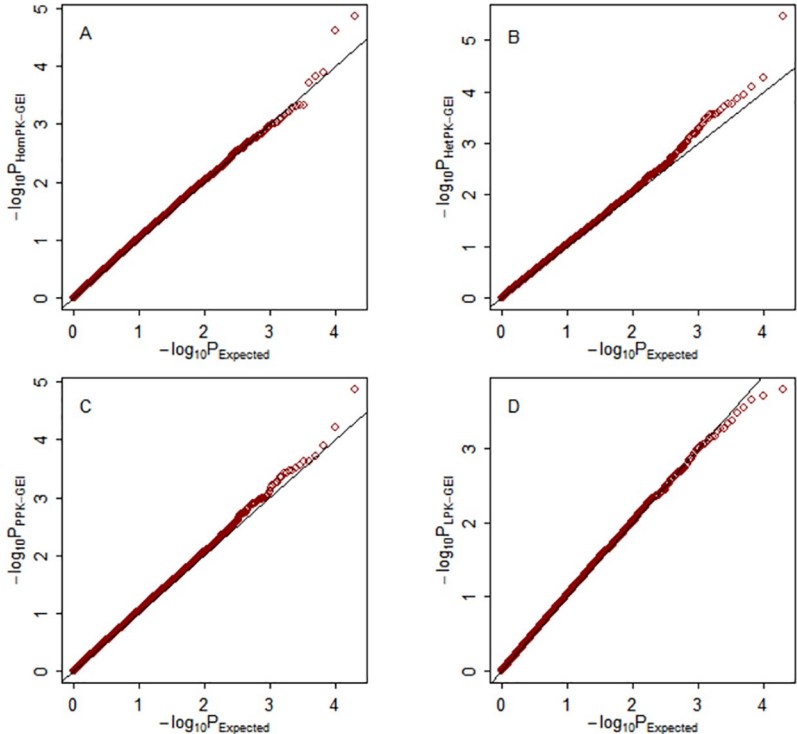

**Fig 3. Q–Q plots of the null distributions under the null hypothesis with strong among-phenotype correlation $\rho = 0.75$.** The horizontal and vertical axes represent the negative $\log_{10}$ of the expected p-values and the negative $\log_{10}$ of the observed p-values, respectively. A: Hom-GEI; B: Het-GEI; C: PPK-GEI; D: LPK-GEI.

is because the phenotypes were simulated based on heterogeneous interaction effects, violating the assumption that Hom-GEI is based upon. Because the GEI effects of a variant on multiple phenotypes can hardly be homogeneous in reality, Hom-GEI may not be a good choice for real data analysis. Therefore, we choose Het-GEI and PPK-GEI for our genome-wide interaction analysis in UK Biobank.

With the proportion of causal rare variant $\theta = 0.1$ and the among-phenotype correlation $\rho = 0.5$, we compared the power of our tests with the TOWGE-FCT under different numbers of phenotypes associated with the interactions, the results are shown in Fig 7. Because TOWGE-FCT is a permutation based method, which is computationally very expensive, the power results were evaluated at the significance level of 0.05. As can be observed from Fig 7 that all the five tests can provide enhanced power as more phenotypes associated with the interactions. Het-GEI, PPK-GEI and LPK-GEI tests have higher power than TOWGE-FCT, however, Hom-GEI has lower power than TOWGE-FCT. We can also see that Het-GEI and PPK-GEI tests outperform the other tests, further indicating Het-GEI and PPK-GEI to be two powerful tests. For instance, when only the first two phenotypes are associated with the interactions, the power values for Hom-GEI, Het-GEI, PPK-GEI, LPK-GEI and TOWGE-FCT are 0.177, 0.507, 0.517, 0.403 and 0.304, respectively.

## Gene-Hb interaction analysis of blood pressure phenotypes in UK Biobank

UK Biobank is a prospective study that recruited approximately 500,000 volunteers aged 40 and 69 years in the United Kingdom and collected extensive genetic and phenotypic data [25, 26]. We used the whole-exome sequencing data released by UK Biobank with a total of

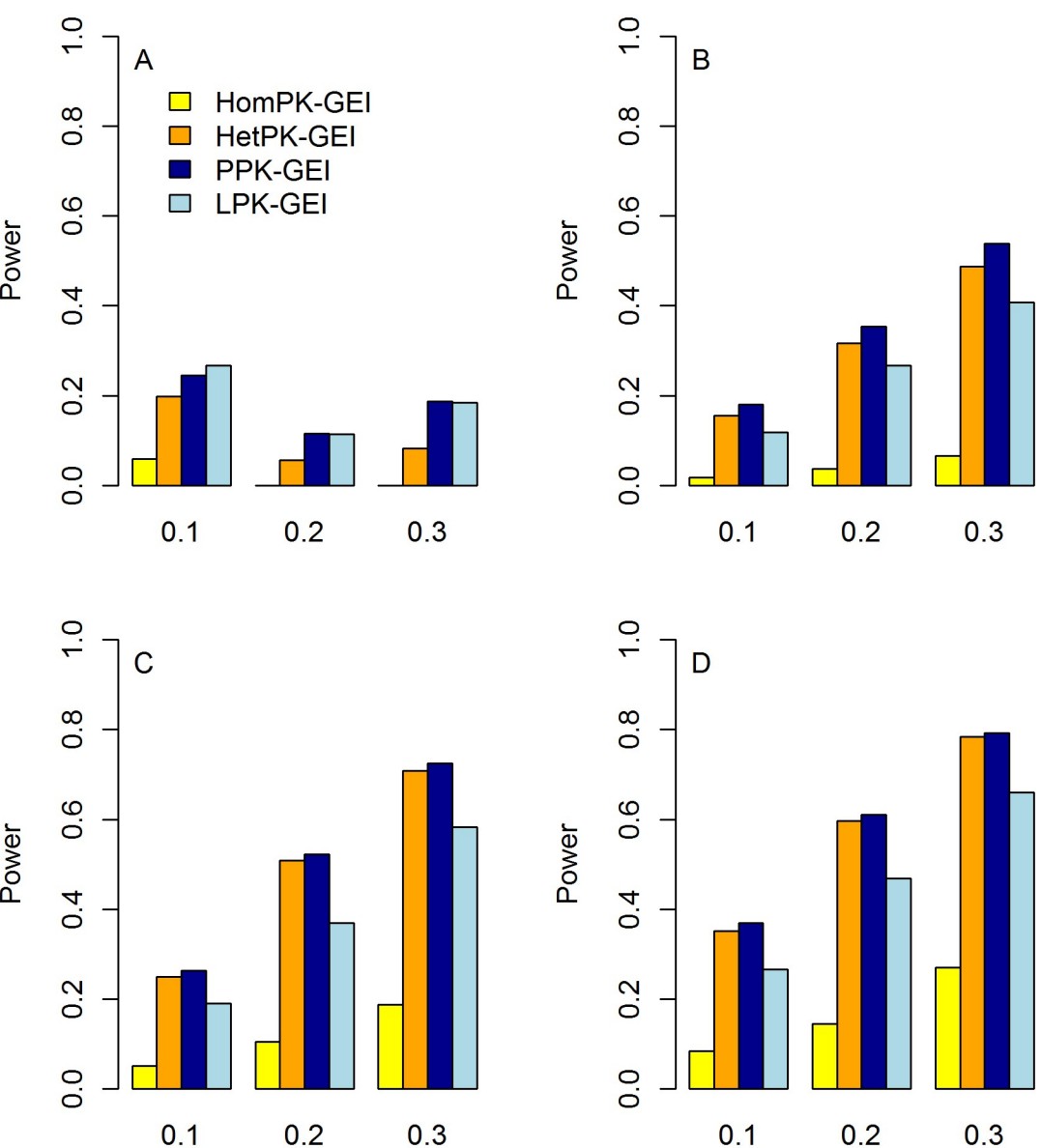

**Fig 4. Statistical power of Hom-GEI, Het-GEI, PPK-GEI and LPK-GEI under weak among-phenotype correlation $\rho$ = 0.25.**
The horizontal and vertical axes represent the proportion of causal rare variants and the statistical power, respectively. A: Power when only the first phenotype is associated with interactions; B: Power when the first two phenotypes are associated with interactions; C: Power when the first three phenotypes are associated with interactions; D: Power when all four phenotypes are associated with interactions.

200,643 samples. Individuals who had withdrawn and one member in each pair with kinship larger than 0.25 measured via KING [27] were removed. We considered SBP and DBP as the blood pressure (BP) phenotypes and Hb as the environmental factor, which is known to be associated with both SBP and DBP [17, 18]. Covariates included age, age$^2$, sex, BMI and 20 principal components to adjust for population stratification. SBP and DBP averaged over multiple measurements at baseline were used. For individuals taking BP-lowering medications, 10 mm Hg and 5 mm Hg were added to the SBP and DBP, respectively [28, 29]. Phenotypes and covariates located 5 standard deviations away from their respective means were defined as

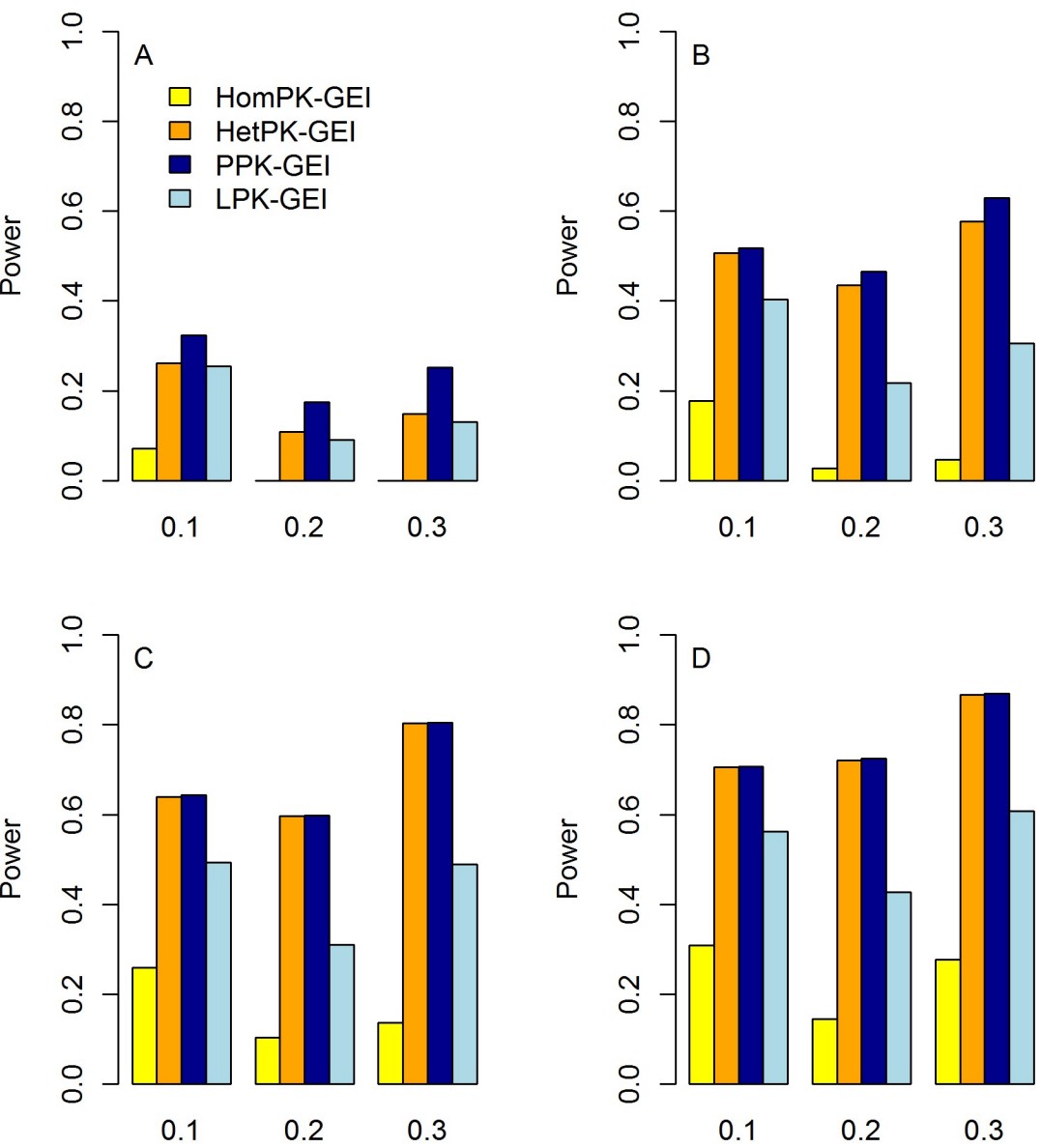

**Fig 5. Statistical power of Hom-GEI, Het-GEI, PPK-GEI and LPK-GEI under moderate among-phenotype correlation $\rho$ = 0.5.** The horizontal and vertical axes represent the proportion of causal rare variants and the statistical power, respectively. A: Power when only the first phenotype is associated with the interactions; B: Power when the first two phenotypes are associated with the interactions; C: Power when the first three phenotypes are associated with the interactions; D: Power when all four phenotypes are associated with the interactions.

outliers. Outliers or individuals with missing phenotypes or missing covariates were removed. As a result, 157,514 individuals, including 71,501 males (45.4% males) and 86,013 females (54.6% females), were included in our gene-Hb interaction analyses. BP phenotypes, age, BMI and Hb were standardized before the analysis.

We carried out genome-wide analysis on 18,101 genes from 22 autosomal chromosomes. Variants in the genotype dataset were annotated via VEP [30]. We restricted to variants annotated as stop_loss, missense_variant, start_lose, splice_donor_variant, inframe_deletion, frameshift_variant, splice_acceptor_variant, stop_gained or inframe_insertion with PolyPhen

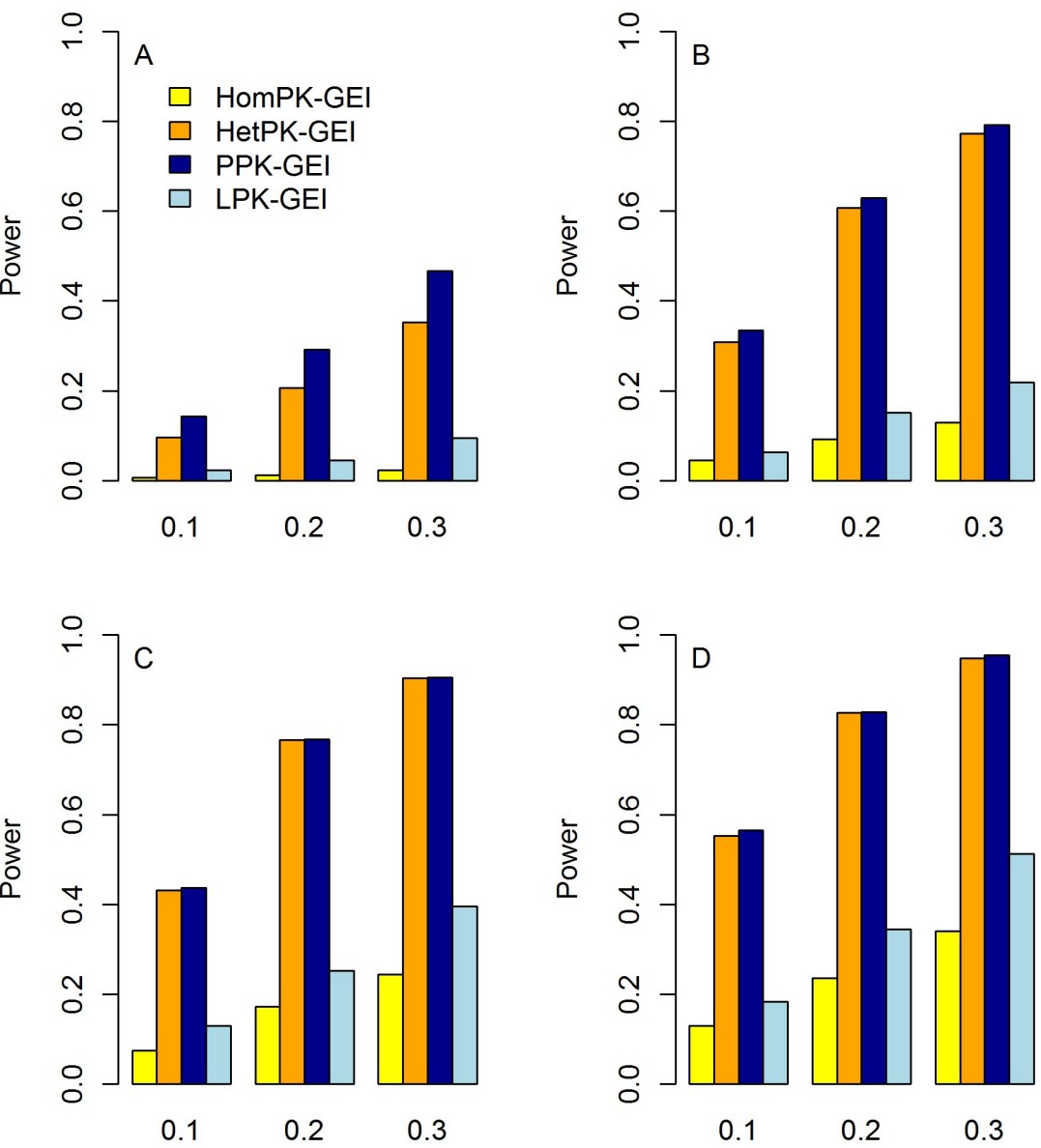

**Fig 6. Statistical power of Hom-GEI, Het-GEI, PPK-GEI and LPK-GEI under strong among-phenotype correlation** $\rho$ = **0.75.** The horizontal and vertical axes represent the proportion of causal rare variants and the statistical power, respectively. A: Power when only the first phenotype is associated with the interactions; B: Power when the first two phenotypes are associated with the interactions; C: Power when the first three phenotypes are associated with the interactions; D: Power when all four phenotypes are associated with the interactions.

scores larger than 0.15 and Sift scores less than 0.05 [31]. Those variants with MAFs less than 3% were extracted via PLINK [32]. Genotypes were further transformed into numeric values using fcGENE [33].

For each of the 18,101 genes, we performed Het-GEI and PPK-GEI analysis of the gene-Hb interactions on SBP and DBP phenotypes. Manhattan plots of p-values from the two tests are presented in Fig 8, and QQ plots of the Het-GEI and PPK-GEI tests are shown in Fig 9. With a genome-wide significance level of $2.5 \times 10^{-6}$, only *LEUTX* is significant according to the Het-GEI test (p-value = $2.2 \times 10^{-6}$), and its p-value according to the PPK-GEI test is $7.43 \times 10^{-6}$. If we

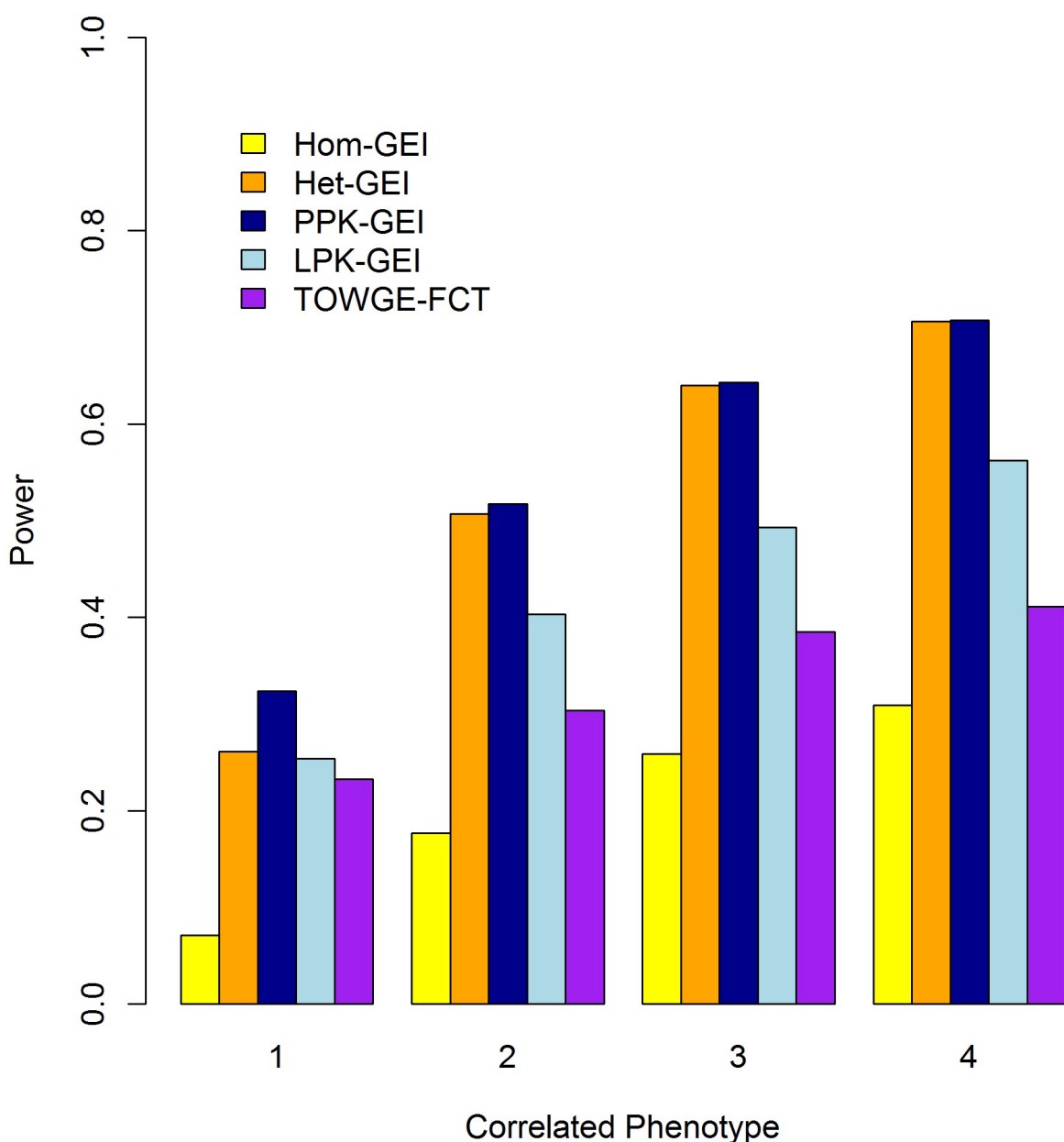

**Fig 7. Statistical power of Hom-GEI, Het-GEI, PPK-GEI, LPK-GEI and TOWGE-FCT with different correlated phenotypes, $\rho = 0.5$, $\theta = 0.1$.** The horizontal and vertical axes represent the number of phenotypes associated with interactions and the statistical power, respectively. The significance level is 0.05.

consider a suggestive significance level at $1 \times 10^{-4}$, twelve genes passed the threshold, whose details are presented in Table 1.

Recently, Surendran et al. reported 22 genes associated with SBP or DBP in a meta-analysis of 1.3 million samples from multiple cohorts, including UK Biobank, the Million Veterans Program and deCODE [19]. For the 22 genes, we looked up our genome-wide results for the possible interactions with Hb. For comparison, we also conducted single-phenotype GEI tests using the INT-FIX function from the rareGE R package [16] for the two BP phenotypes

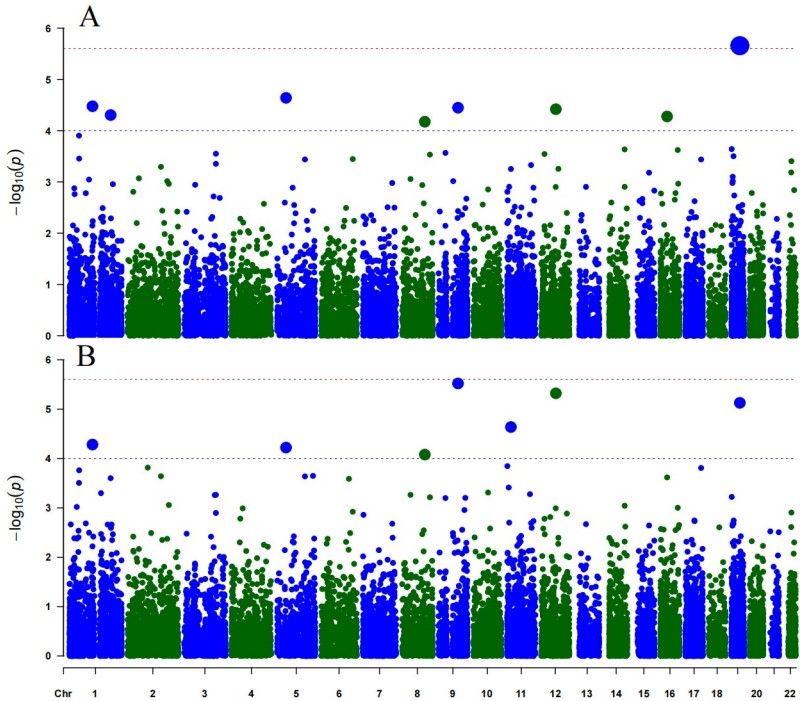

**Fig 8. Manhattan plot of genome-wide multiphenotype analysis of gene-Hb interactions in BPs.** The horizontal and vertical axes represent the genomic position and the negative $\log_{10}$ of the p-values, respectively. A: Het-GEI; B: PPK-GEI.

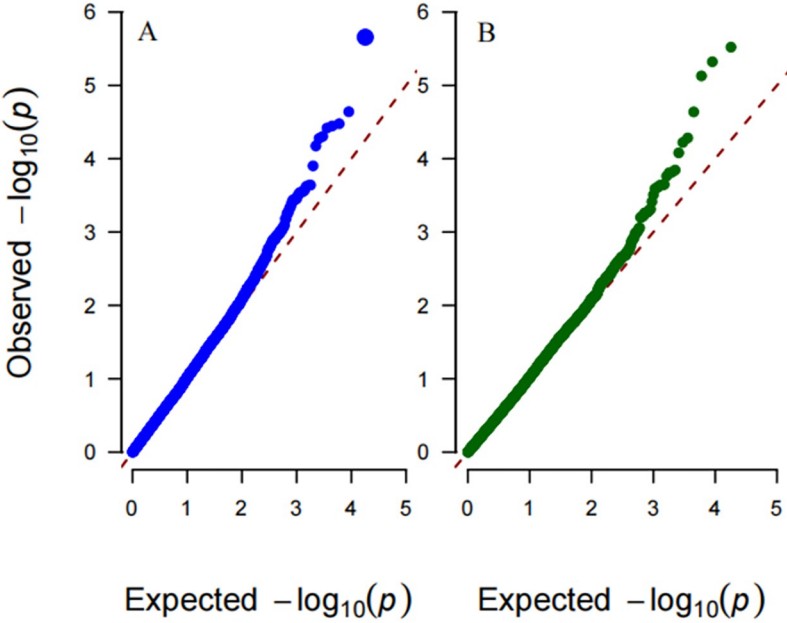

**Fig 9. Q–Q plots of genome-wide multiphenotype GEI analysis of gene-Hb interactions in BPs.** The horizontal and vertical axes represent the negative $\log_{10}$ of the expected p-values and the negative $\log_{10}$ of the observed p-values, respectively. A: Het-GEI; B: PPK-GEI.

**Table 1. Genes showing suggestive evidence (p-values $< 1 \times 10^{-4}$) of gene-Hb interactions in the Het-GEI or PPK-GEI tests.**

| Gene | Chr | Pos | RV Num | P value | |
|------|-----|-----|--------|---------|---|
| | | | | Het-GEI | PPK-GEI |
| ALX3 | 1 | 110060374: 110070700 | 27 | $3.34 \times 10^{-5}$ | $5.21 \times 10^{-5}$ |
| CFHR1 | 1 | 196819730: 196832189 | 19 | $4.96 \times 10^{-5}$ | $2.50 \times 10^{-4}$ |
| ARID5A | 2 | 96536726: 96552634 | 16 | $9.60 \times 10^{-2}$ | $1.53 \times 10^{-4}$ |
| OXCT1 | 5 | 41730064: 41870689 | 18 | $2.29 \times 10^{-5}$ | $5.99 \times 10^{-5}$ |
| EMC2 | 8 | 108443623: 108486907 | 11 | $6.69 \times 10^{-5}$ | $8.32 \times 10^{-5}$ |
| COQ4 | 9 | 128322511: 128334072 | 36 | $2.12 \times 10^{-1}$ | $6.26 \times 10^{-4}$ |
| FAM120AOS | 9 | 93446499: 93453592 | 5 | $3.57 \times 10^{-5}$ | $3.01 \times 10^{-6}$ |
| NCR3LG1 | 11 | 17351761: 17377321 | 6 | $5.59 \times 10^{-4}$ | $2.30 \times 10^{-5}$ |
| OR2D3 | 11 | 6921001: 6921994 | 15 | $1.36 \times 10^{-1}$ | $3.86 \times 10^{-4}$ |
| CPSF6 | 12 | 69239536: 69274358 | 4 | $3.81 \times 10^{-5}$ | $4.76 \times 10^{-6}$ |
| PYDC1 | 16 | 31215961: 31217074 | 2 | $5.29 \times 10^{-5}$ | $2.41 \times 10^{-4}$ |
| LEUTX | 19 | 39776593: 39786135 | 15 | $2.20 \times 10^{-6}$ | $7.43 \times 10^{-6}$ |

Abbreviations are as follows: Chr, chromosome. Pos, position. RV Num, number of rare variants.

separately. The number of rare variants involved in the analysis and p-values from the multi-phenotype and single-phenotype GEI tests are provided in Table 2.

There are no significant results after correcting the multiple testing. At the nominal significance level of 0.05, only one gene, *MYO1C*, shows interactions with Hb for BP phenotypes by the PPK-GEI test (p-value = 0.038). With the single-phenotype GEI test, *NOS3* has a p-value of 0.026 for DBP, and *MYO1C* has p-values of 0.018 and 0.011 for SBP and DBP, respectively.

## Discussion

In this paper, we propose four statistical tests, Hom-GEI, Het-GEI, PPK-GEI and LPK-GEI, to test GEI effects with rare variants for multiple correlated quantitative phenotypes. Through simulation studies, the statistical power of the tests was investigated in terms of the proportion of causal variants, the number of phenotypes associated with interactions and the correlation strength among phenotypes. Simulation results show that all tests demonstrate improved statistical power when the proportion of causal variants or the number of associated phenotypes increases. Hom-GEI, Het-GEI and PPK-GEI benefit from correlation among phenotypes; however, the LPK-GEI test suffers power loss, especially when correlation among phenotypes is strong. This is because LPK-GEI directly combines statistics from single-phenotype GEI tests and ignores phenotype dependence. In addition, among almost all of the considered scenarios, Het-GEI and PPK-GEI have almost the same power and outperform the other two tests. Hom-GEI shows the poorest power due to its unrealistic assumption. In summary, Het-GEI and PPK-GEI are two powerful tests for investigating GEI with multiple quantitative phenotypes.

We applied Het-GEI and PPK-GEI in the genome-wide analysis of SBP and DBP in order to detect possible gene-Hb interactions in UK Biobank. We analyzed 18,101 genes and identified *LEUTX* to be associated with BP phenotypes through its interaction with Hb via the Het-GEI test. At the suggestive significance level, twelve genes were identified to be associated with BP phenotypes through their interactions with Hb. *LEUTX* was previously reported to play a central role in embryo genome activation [34] whose role in BP regulation is unclear. Recent study of rare variants suggests that BP-associated variants are enriched in active chromatin regions of fetal tissue and potentially link fetal development to BP regulation in later life [19].

**Table 2. Multiphenotypic analyses and single-phenotype analyses of gene-Hb interactions in BP-associated genes.**

| Gene | Chr | Pos | RV Num | P value | | | |
|------|-----|-----|--------|---------|---------|---------|---------|
| | | | | Het-GEI | PPK-GEI | INT-FIX (SBP) | INT-FIX (DBP) |
| NPR1 | 1 | 153678687: 153693992 | 75 | 0.459 | 0.528 | 0.688 | 0.291 |
| AGT | 1 | 230702522: 230714590 | 46 | 0.556 | 0.497 | 0.599 | 0.213 |
| PHC3 | 3 | 170087579: 170181749 | 40 | 0.630 | 0.371 | 0.264 | 0.157 |
| NR3C2 | 4 | 148078763: 148442520 | 17 | 0.922 | 0.690 | 0.542 | 0.240 |
| SLC39A8 | 4 | 102251040: 102345498 | 23 | 0.766 | 0.522 | 0.296 | 0.289 |
| ENPEP | 4 | 110476072: 110563337 | 74 | 0.663 | 0.785 | 0.510 | 0.912 |
| NOS3 | 7 | 150991055: 151014599 | 126 | 0.183 | 0.065 | 0.129 | 0.026 |
| ZFAT | 8 | 134477787: 134713049 | 67 | 0.091 | 0.161 | 0.353 | 0.309 |
| DBH | 9 | 133636362: 133659344 | 81 | 0.828 | 0.627 | 0.371 | 0.339 |
| PLCE1 | 10 | 93993988: 94328391 | 73 | 0.269 | 0.343 | 0.495 | 0.269 |
| PLCB3 | 11 | 64251522: 64269452 | 85 | 0.227 | 0.344 | 0.241 | 0.792 |
| ALKBH8 | 11 | 107502726: 107565735 | 41 | 0.924 | 0.926 | 0.644 | 0.871 |
| PDE3B | 11 | 14643722: 14872058 | 81 | 0.336 | 0.542 | 0.780 | 0.500 |
| PDE3A | 12 | 20369244: 20684107 | 26 | 0.205 | 0.304 | 0.440 | 0.400 |
| RAPGEF3 | 12 | 47734669: 47759106 | 63 | 0.051 | 0.103 | 0.106 | 0.888 |
| HDAC7 | 12 | 47782723: 47819980 | 38 | 0.402 | 0.470 | 0.673 | 0.336 |
| HEATR5A | 14 | 31291787: 31420582 | 129 | 0.740 | 0.468 | 0.165 | 0.382 |
| SOS2 | 14 | 50117127: 50231381 | 60 | 0.170 | 0.491 | 0.647 | 0.981 |
| SLC9A3R2 | 16 | 2026867: 2039026 | 47 | 0.968 | 0.893 | 0.685 | 0.594 |
| MYO1C | 17 | 1464185: 1492707 | 165 | 0.468 | 0.038 | 0.018 | 0.011 |
| GATA5 | 20 | 62463496: 62475970 | 77 | 0.610 | 0.704 | 0.756 | 0.475 |
| FOXS1 | 20 | 31844299: 31845617 | 44 | 0.563 | 0.553 | 0.285 | 0.592 |

Abbreviations are as follows: Chr, chromosome. RV Num, number of rare variants. Pos, position. SBP, systolic blood pressure. DBP, diastolic blood pressure.

Thus, *LEUTX* might be associated with the BP phenotypes in a similar manner. In the analysis of 454,787 UK Biobank participants, genetic main effect of *LEUTX* was not associated with BP phenotypes at the nominal significance level of 0.05 [35]. However, our tests identified *LEUTX* to be interacted with Hb in BP phenotypes at the genome-wide significance level of $2.5 \times 10^{-6}$ in the sample of 157,514 UK Biobank participants. We also conducted a single-phenotype GEI test and multiphenotype GEI tests to evaluate the gene-Hb interactions for 22 genes that were previously reported to be associated with SBP or DBP in a meta-analysis of genetic main effects. *MYO1C* shows nominal significance by the Het-GEI test. *NOS3* shows nominal significance in DBP and *MYO1C* in SBP and DBP by the single-phenotype GEI test.

Our proposed multiphenotype GEI tests are an extension of the single-phenotype GEI test in rareGE [16]. The tests retain the desirable properties of the single-phenotype GEI test, which allows for adjusting covariates and is powerful when the GEI effects of variants act in different directions on phenotypes. Our proposed multiphenotype GEI tests, except for Hom-GEI, are more powerful than the existing multiphenotype GEI test TOWGE-FCT. Moreover, our methods are computationally less expensive. This is because that TOWGE-FCT employs permutations to evaluate p value for each transferred phenotype, however, our tests are based on asymptotic distributions and p values can be computed analytically.

Our proposed multiphenotype GEI tests have the following limitations. First, our tests have specific assumptions on the correlation structure for the GEI effects among multiple phenotypes, violating the assumption would lead to a substantial loss of power. Second, our tests

assume individuals to be unrelated. Familiar correlation is not considered, and related samples must be excluded, which may substantially reduce the sample size and result in a loss of power. Third, our proposed GEI tests are for rare variants only, while both common and rare variants may contribute to complex diseases [36–38]. In future work, we plan to consider a unified test to address these issues. Finally, while we identified *LEUTX* to interact with Hb in BP phenotypes, the result lacks independent validation. Thus, it should be considered as being preliminary and further experiments are necessary.

## Conclusion

In this paper, we modeled the correlation among the GEI effects of a variant on multiple phenotypes by using four kernels. Based on these kernels, we proposed four multiphenotype GEI tests for rare variants. The four tests retain the desirable properties of the single-phenotype GEI test and provide enhanced statistical power by analyzing multiple phenotypes simultaneously. We applied Het-GEI and PPK-GEI to test gene-Hb interactions for 18,101 genes in SBP and DBP in UK Biobank. *LEUTX* was associated with BP phenotypes through the interaction with Hb via the Het-GEI test. At the suggestive significance level, twelve genes were reported. Our proposed tests can be readily used to test GEIs in a variety of correlated phenotypes and hopefully contribute to the genetic studies of complex diseases.

## Author Contributions

**Conceptualization:** Xiaoqin Jin, Gang Shi.

**Formal analysis:** Xiaoqin Jin.

**Funding acquisition:** Gang Shi.

**Methodology:** Xiaoqin Jin, Gang Shi.

**Software:** Xiaoqin Jin.

**Supervision:** Xiaoqin Jin, Gang Shi.

**Validation:** Xiaoqin Jin.

**Writing – original draft:** Xiaoqin Jin.

**Writing – review & editing:** Gang Shi.

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
