## [Decision Letter · Decision Letter 0]

15 Jun 2022

PONE-D-22-11234Kernel-based gene–environment interaction tests for rare variants with multiple quantitative phenotypes PLOS ONE

Dear Dr. Shi,

Thank you for submitting your manuscript to PLOS ONE. After careful consideration, we feel that it has merit but does not fully meet PLOS ONE’s publication criteria as it currently stands. Therefore, we invite you to submit a revised version of the manuscript that addresses the points raised during the review process.

We look forward to receiving your revised manuscript.

Kind regards,

Momiao Xiong

Academic Editor

PLOS ONE

Journal Requirements:

"This work was supported by the national Thousand Youth Talents Plan.

Grant recipient: Gang Shi"

Reviewers' comments:

Reviewer's Responses to Questions

**Comments to the Author**

1. Is the manuscript technically sound, and do the data support the conclusions?

Reviewer #1: Yes

2. Has the statistical analysis been performed appropriately and rigorously? 

Reviewer #1: Yes

3. Have the authors made all data underlying the findings in their manuscript fully available?

Reviewer #1: Yes

4. Is the manuscript presented in an intelligible fashion and written in standard English?

Reviewer #1: Yes

5. Review Comments to the Author

Reviewer #1: This is a very clear and well-organized paper that discusses four multiphenotype methods for testing GEIs with rare variants, which is a great contribution to the field of GEI pleiotropy where most literature focuses on common variants testing for multiple phenotypes.

I have a few comments below:

1. Have you considered comparing your four proposed methods with current existing methods in GEI pleiotropy field via simulations? Also, can you discuss what are the advantages of the proposed models and limitations of the existing methods?

2. In the UK Biobank data application, there are several genes identified by your proposed models to have interactions. Are these genes confirmed in past literature to show correlations with BP phenotypes? Have you tried anyway to validate the results as well as your models?

6. PLOS authors have the option to publish the peer review history of their article (what does this mean?). If published, this will include your full peer review and any attached files.

Reviewer #1: No

---

## [Author Response · Author response to Decision Letter 0]

6 Jul 2022

Respond to Editors:

We are herein resubmitting our revised manuscript titled “Kernel-Based Gene–Environment Interaction Tests for Rare Variants with Multiple Quantitative Phenotypes” (Manuscript Number: PONE-D-22-11234) to PLoS One. 

We would like to thank you for giving us the opportunity to revise our manuscript and respond to the comments raised by the reviewer. We also want to thank the reviewer for his detailed comments and constructive suggestions, which help improving the quality of this paper substantially. The manuscript has been revised according to the comments of the reviewer and a point-by-point response to the reviewer’s comments is enclosed. 

The resubmitted manuscript has been formatted according to the style of PLoS One. We have checked our reference list and added four references in the revised manuscript, the four references were cited in new contents that have added in the revised manuscript according to the reviewer’s comments. ORCID ID has been linked to our Editorial Manager account. Besides, we have added “Author Contributions” in our revised manuscript. Any changes made to our original manuscript have been highlighted in our revised manuscript. The “Funding” statement and “Data Availability Statement” are corrected as follows:

Funding

This work was supported by the national Thousand Youth Talents Plan to GS. The funders had no role in study design, data collection and analysis, decision to publish, or preparation of the manuscript.

Data Availability Statement

This research has been conducted using the UK Biobank Resource under Application Number 44080. The genetic and phenotype datasets are available via the UK Biobank data access process. More details are available at http://www.ukbiobank.ac.uk/register-apply/.

Respond to Reviewer:

1.Have you considered comparing your four proposed methods with current existing methods in GEI pleiotropy field via simulations? Also, can you discuss what are the advantages of the proposed models and limitations of the existing methods?

Response: Thank you for the suggestion! We conducted an extensive literature search and found only one method available for testing GEI with rare variants and multiple phenotypes (Zhang et al. 2019). The method consists of three steps: remove correlation among multiple phenotypes by using principal component analysis or other linear transformations; obtain p value for each transformed phenotype by testing the effects of an optimally weighted combination of gene-environment interactions for rare variants (TOW-GE) (Zhao et al. 2020); employ Fisher’s combination test (FCT) to combine the p values of multiple phenotypes. We denote the method as TOWGE-FCT in this paper. It can be expected that the degree of freedom of TOWGE-FCT would become larger with the increasing number of phenotypes, which might limit statistical power of the test.

In the revised manuscript, we have added simulations and compared our proposed four multiphenotype GEI tests with TOWGE-FCT. The results showed that all the five tests have enhanced power as the number of phenotypes associated with interactions increases, our proposed Het-GEI, PPK-GEI and LPK-GEI had larger power than TOWGE-FCT, but Hom-GEI had smaller power than TOWGE-FCT. Moreover, TOWGE-FCT uses a permutation test to estimate p value for each phenotype, which is very time consuming. In comparison, our proposed four tests have asymptotic distributions and p values can be computed analytically. Thus, our proposed tests have much smaller computational cost. We also added discussion on the advantages of our proposed tests and limitations of TOWGE-FCT in the revised manuscript. 

Zhang J, Sha Q, Hao H, Zhang S, Gao XR, Wang X. Test Gene-Environment Interactions for Multiple Traits in Sequencing Association Studies. Hum Hered. 2019;84(4-5):170-196. https://doi.org/10.1159/000506008 PMID: 32417835

Zhao Z, Zhang J, Sha Q, Hao H. Testing gene-environment interactions for rare and/or common variants in sequencing association studies. PLoS One. 2020;15(3):e0229217. https://doi.org/10.1371/journal.pone.0229217 PMID: 32155162

2.In the UK Biobank data application, there are several genes identified by your proposed models to have interactions. Are these genes confirmed in past literature to show correlations with BP phenotypes? Have you tried anyway to validate the results as well as your models?

Response: Thank you for the suggestion! In our work, we identified three genes that have interactions with Hb in BP phenotypes. For NOS3 and MYO1C genes, they have been reported to be correlated with BP phenotypes in (Surendran et al. 2020). 

For LEUTX gene, in an analysis of 454,787 UK Biobank participants, LEUTX was not associated with the BP phenotypes at the nominal significance level of 0.05 (Backman et al. 2021). While our proposed tests identified LEUTX that shows interaction with Hb in BP phenotypes, the result lacks independent validation. Thus, it should be considered as being preliminary and further experiments are necessary. We acknowledged the limitation of the result in the revised manuscript.

Surendran P, Feofanova EV, Lahrouchi N, Ntalla I, Karthikeyan S, Cook J, et al. Discovery of rare variants associated with blood pressure regulation through meta-analysis of 1.3 million individuals. Nat Genet. 2020;52(12):1314-1332. https://doi.org/10.1038/s41588-020-00713-x PMID: 33230300 

Backman JD, Li AH, Marcketta A, Sun D, Mbatchou J, Kessler MD, et al. Exome sequencing and analysis of 454,787 UK Biobank participants. Nature. 2021;599(7886):628-634. https://doi.org/10.1038/s41586-021-04103-z PMID: 34662886

---

## [Decision Letter · Decision Letter 1]

26 Sep 2022

Kernel-based gene–environment interaction tests for rare variants with multiple quantitative phenotypes

PONE-D-22-11234R1

Dear Dr. Shi,

We’re pleased to inform you that your manuscript has been judged scientifically suitable for publication and will be formally accepted for publication once it meets all outstanding technical requirements.

Kind regards,

Mehdi Rahimi, Ph.D.

Academic Editor

PLOS ONE

Additional Editor Comments (optional):

Reviewers' comments:

Reviewer's Responses to Questions

**Comments to the Author**

1. If the authors have adequately addressed your comments raised in a previous round of review and you feel that this manuscript is now acceptable for publication, you may indicate that here to bypass the “Comments to the Author” section, enter your conflict of interest statement in the “Confidential to Editor” section, and submit your "Accept" recommendation.

Reviewer #1: All comments have been addressed

2. Is the manuscript technically sound, and do the data support the conclusions?

Reviewer #1: Yes

3. Has the statistical analysis been performed appropriately and rigorously? 

Reviewer #1: Yes

4. Have the authors made all data underlying the findings in their manuscript fully available?

Reviewer #1: Yes

5. Is the manuscript presented in an intelligible fashion and written in standard English?

Reviewer #1: Yes

6. Review Comments to the Author

Reviewer #1: (No Response)

7. PLOS authors have the option to publish the peer review history of their article (what does this mean?). If published, this will include your full peer review and any attached files.

Reviewer #1: No

---

## [Editor Report · Acceptance letter]

30 Sep 2022

PONE-D-22-11234R1 

Kernel-based gene–environment interaction tests for rare variants with multiple quantitative phenotypes 

Dear Dr. Shi:

I'm pleased to inform you that your manuscript has been deemed suitable for publication in PLOS ONE. Congratulations! Your manuscript is now with our production department. 

Kind regards, 

on behalf of

Dr. Mehdi Rahimi 

Academic Editor

PLOS ONE